# Hepatic Mitochondrial Dysfunction and Risk of Liver Disease in an Ovine Model of “PCOS Males”

**DOI:** 10.3390/biomedicines10061291

**Published:** 2022-05-31

**Authors:** Katarzyna J. Siemienowicz, Panagiotis Filis, Jennifer Thomas, Paul A. Fowler, W. Colin Duncan, Mick T. Rae

**Affiliations:** 1School of Applied Science, Edinburgh Napier University, Edinburgh EH11 4BN, UK; jennifer.thomas@napier.ac.uk (J.T.); m.rae@napier.ac.uk (M.T.R.); 2MRC Centre for Reproductive Health, The University of Edinburgh, Edinburgh EH16 4TJ, UK; w.c.duncan@ed.ac.uk; 3Institute of Medical Sciences, School of Medicine, Medical Sciences & Nutrition, University of Aberdeen, Aberdeen AB25 2ZD, UK; pfilis@abdn.ac.uk (P.F.); p.a.fowler@abdn.ac.uk (P.A.F.)

**Keywords:** male PCOS, NAFLD, NASH, androgens, prenatal programming, mitochondrial dysfunction, hepatic cholesterol, oxidative phosphorylation, liver fibrosis

## Abstract

First-degree male relatives of polycystic ovary syndrome (PCOS) sufferers can develop metabolic abnormalities evidenced by elevated circulating cholesterol and triglycerides, suggestive of a male PCOS equivalent. Similarly, male sheep overexposed to excess androgens in fetal life develop dyslipidaemia in adolescence. Dyslipidaemia, altered lipid metabolism, and dysfunctional hepatic mitochondria are associated with the development of non-alcoholic liver disease (NAFLD). We therefore dissected hepatic mitochondrial function and lipid metabolism in adolescent prenatally androgenized (PA) males from an ovine model of PCOS. Testosterone was directly administered to male ovine fetuses to create prenatal androgenic overexposure. Liver RNA sequencing and proteomics occurred at 6 months of age. Hepatic lipids, glycogen, ATP, reactive oxygen species (ROS), DNA damage, and collagen were assessed. Adolescent PA males had an increased accumulation of hepatic cholesterol and glycogen, together with perturbed glucose and fatty acid metabolism, mitochondrial dysfunction, with altered mitochondrial transport, decreased oxidative phosphorylation and ATP synthesis, and impaired mitophagy. Mitochondrial dysfunction in PA males was associated with increased hepatic ROS level and signs of early liver fibrosis, with clinical relevance to NAFLD progression. We conclude that excess in utero androgen exposure in male fetuses leads to a PCOS-like metabolic phenotype with dysregulated mitochondrial function and likely lifelong health sequelae.

## 1. Introduction

Polycystic ovary syndrome (PCOS) is the most common hormonal disorder affecting over 10% of premenopausal women, characterized by hyperandrogenaemia and reproductive and metabolic abnormalities [1]. PCOS is associated with increased incidence of insulin resistance, dyslipidaemia, non-alcoholic fatty liver disease (NAFLD), and obesity [1,2]. PCOS shows familial clustering and appears to be transmitted between generations [3,4]. Preclinical and clinical studies have highlighted that increased prenatal androgen exposure is associated with development of a PCOS phenotype in adult life [5,6,7], and the daughters of women with PCOS have increased cord blood testosterone [8], longer anogenital distance [9], and increased sebum production [10], indicative of increased in utero androgen exposure.

Recently a male phenotype of the metabolic arm of PCOS was recognised. First-degree male relatives of women with PCOS also develop metabolic, cardiovascular, and hormonal abnormalities [11,12,13,14,15,16,17,18,19], suggesting the existence of a male PCOS equivalent [20,21,22]. Indeed, this recognition of a “male PCOS” metabolic phenotype has led to the suggestion of recognition of PCOS in both sexes as “Metabolic Reproductive Syndrome” [22]. Elevated circulating cholesterol and triglycerides are early markers of metabolic dysfunction in sons of women with PCOS [16,17]. We recently demonstrated that in utero androgen excess in an ovine model of PCOS recreated the metabolic phenotype of males with first degree PCOS relatives [23]. Prenatal androgen excess in male sheep was associated with adolescent hyperinsulinemia, dyslipidaemia, and increased anti-Mullerian hormone concentrations. We also observed that the “PCOS male” had dysregulated hepatic cholesterol homeostasis, a developing intrahepatic cholestasis-like condition, decreased hepatic detoxification potential, and increased risk of hepatic fibrosis in later life [23].

Liver diseases are a serious global health concern [24]. NAFLD is the most common liver disease, affecting both adults and children, and is one of the leading causes for liver transplantation [25,26]. NAFLD is a spectrum of chronic liver diseases varying from elevated hepatic lipid storage in steatosis to increased inflammation and hepatocyte damage in steatohepatitis (NASH), which, in turn, increases the risk of cirrhosis, liver failure, and hepatic carcinoma [27]. NAFLD is often accompanied by co-morbidities such as obesity, hypertension, dyslipidaemia and altered lipid metabolism, insulin resistance or diabetes, and cardiovascular disease, which can further bi-directionally increase the risk of advanced disease and worsen outcomes [25,28,29]. Women with PCOS have an increased risk of NAFLD [2], and androgen excess has been implicated as a potential driver of NAFLD in PCOS [30].

Mitochondria have a central role in energy production, and mitochondrial dysfunction at the cellular level can affect systemic metabolic balance [31]. Dysfunctional hepatic mitochondria contribute to NAFLD development and progression [32,33], and it has been suggested that mitochondrial dysfunction may be a contributing factor in the pathogenesis of PCOS [34]. We therefore aimed to dissect hepatic mitochondrial function and lipid metabolism in adolescent prenatally androgenized (PA) males from an ovine model of PCOS [23]. We report mitochondrial dysfunction as a consequence of male prenatal androgen excess, and we report the outcomes in terms of lipid handling, energy metabolism, and potential liver damage, describing the underpinning mechanisms contributing to such metabolic dysfunction.

## 2. Materials and Methods

### 2.1. Ethics Statement

All studies were approved by the UK Home Office and conducted under approved Project Licence PPL 60/4401, reviewed by The University of Edinburgh Animal Research Ethics Committee.

### 2.2. Animals and Tissue Collection

A detailed description of animal husbandry, experimental protocols, and tissue collection was previously published [35,36]. In brief, studies were conducted under natural lighting conditions, without environmental manipulations. Scottish Greyface ewes were fed hay ad libitum, and prior to mating, ewes with a body condition score of 2.75–3 were synchronized with Chronogest sponges (flugestone acetate (Intervet Ltd., Milton Keynes, UK)) and Estrumate (cloprostenol) injection (Schering Plough Animal Health, Milton Keynes, UK). Ewes were mated with Texel rams under natural seasonal breeding conditions, and pregnancies were confirmed by ultrasound scanning.

At random, animals were allocated to experimental groups, namely testosterone propionate (PA) or vehicle control treatment. On days 62 and 82 of gestation, pregnant ewes were anesthetized with 10 mg Xylazine (i.m. Rompun; Bayer PLC Animal Health Division, Reading, UK) and 2 mg/kg Ketamine (i.v, Keteset; Fort Dodge Animal Health, Sandwich, UK). Under surgical aseptic conditions, the fetuses were injected into the fetal flank (via ultrasound guidance) with a 20G Quinke spinal needle (BD Biosciences, Wokingham, UK) with testosterone propionate (PA; n = 14; 20 mg in 0.2 mL vegetable oil) or vehicle control (C; n = 14; 0.2 mL vegetable oil) [23]. All pregnant ewes were given prophylactic antibiotics (Streptacare, Animalcare Ltd., York, UK, 1 mL/25 kg) directly after surgical procedure completion and were monitored during recovery; there were no adverse effects from these procedures.

At 3 months of age, the lambs were weaned and fed hay and grass ad libitum. The lambs were sacrificed in adolescence at 6 months of age, in a random order, via barbiturate overdose. Prior to euthanasia, peripheral blood was sampled, followed by intravenous administration of bolus glucose (10 g glucose in 20 mL saline), followed by another blood sample collection 15 min later. Blood was collected into heparinised tubes and plasma stored at −20 °C. Fetal tissue was collected on day 90 of gestation (day 147 is term), as previously described [35,36]. Liver samples (fetal and postnatal) were taken from the same lobe (right posterior) and were fixed and processed or immediately snap-frozen and stored at −80 °C until downstream analysis, as previously reported [35,36].

### 2.3. DNA, RNA, and Protein Extraction

A Qiagen AllPrep kit (Qiagen Ltd., Germantown, MD, USA) was used to extract DNA, RNA, and protein, following manufacturer’s instructions. To control for possible batch effects, all samples were randomized within a larger study set. During RNA purification, on-column DNase I digestion was carried out with RNase-Free DNase set (Qiagen Ltd., Germantown, MD, USA). Concentrations of DNA and RNA were measured by using a NanoDrop 1000 spectrophotometer (Thermo Fisher Scientific, Wilmington, DE, USA), and RNA quality control was determined by using an Agilent Bioanalyser (Agilent, Santa Clara, CA, USA); RNA samples used for downstream analysis had RIN values of >7.5. RNA and protein samples were stored at −80 °C until further analysis.

### 2.4. RNA Sequencing Transcriptomic Analyses and Liver Protein Quantification

The RNA sequencing and hepatic protein quantification were previously described in detail [23]. Illumina A TruSeq Stranded mRNA kit (Illumina, Inc., San Diego, CA, USA) was used to prepare RNA sequencing libraries. Sequencing was carried by using the Illumina NextSeq 500 platform (Illumina, Inc., San Diego, CA, USA). Raw sequencing data can be downloaded from the ArrayExpress database (http://www.ebi.ac.uk/arrayexpress; last accessed 26 May 2022), accession number E-MTAB-8032. EdgeR was used to perform pairwise gene comparisons, with all genes with a count per million value of more than one in six. After removing low-count genes, there were 15,134 genes for analysis. The *p*-values were adjusted by using the Benjamini–Hochberg method [37], with a false discovery rate (FDR) set at q < 0.05.

A Q Exactive Plus hybrid quadrupole Orbitrap mass spectrometer fitted with an EASY-Spray nano-ESI source (Thermo Fisher Scientific, Wilmington, DE, USA) was used to identify and quantify hepatic proteins. Limma was used to perform pairwise protein comparisons for those proteins that yielded normalised intensities in at least 75% of the compared samples. The *p*-values were adjusted by using the Benjamini–Hochberg method. Mass spectrometry proteomics data were added to the ProteomeXchange Consortium via the PRIDE140 partner repository, with the dataset identifier PXD014050.

### 2.5. Bioinformatic Analysis

IPA (Ingenuity Pathway Analysis, version 01-20-04, Qiagen, Redwood City, CA, USA) was used to perform initial data screens to identify affected pathways and potential disease states associated with the differentially expressed genes (DEGs) and differentially expressed proteins (DEPs). Pre-filtering of data uploaded to IPA was based upon FDR adjusted significance, using a q < 0.05 as a cutoff. Related genes were then manually identified to further probe the extent and depth of the legacy left by prenatal androgen excess and increase mechanistic understanding. In proteomics analysis, due to smaller numbers of proteins identified, to generate meaningful bioinformatics analyses, the stringency of data input was relaxed by using nominal *p*-values of <0.05. IPA *p*-values of <0.05, calculated with right-tailed Fisher’s exact test, provided information of potential links of significant DEGs and DEPs with biological pathways or diseases, and Z score (where available) delivered information of directional change with regards to such biological pathways.

### 2.6. Quantitative RT-PCR

Complimentary DNA (cDNA) was synthesized by using Reverse Transcription Premix 2 (Primerdesign Ltd., Eastleigh, UK), according to the manufacturer’s protocol. SYBR Green quantitative PCR was prepared with 10 µL Precision Plus qPCR Master Mix (Primerdesign Ltd., Eastleigh, UK), 1 µL primer pairs (0.5 µM), 5 µL RNA (25 ng), and 4 µL nuclease free water. Reactions were carried out in duplicate, using a StepOne Plus platform (Applied Biosystems, Waltham, MA, USA). A melting curve analysis revealed a single amplicon in all cases. Genorm analysis (PrimerDesign Ltd., Eastleigh, UK) was performed in order to identify the most stable housekeeping genes, and *RPL19* was selected as the normalization reference. RT-ve and a template negative reaction were used as negative controls. Primers were designed with the Primer3Plus online software, were synthesised by Eurofins MWG Operon (Ebersberg, Germany), and were validated in house before use (Appendix A). The transcript abundance of target gene relative to the housekeeping genes was quantified by using the 2^−ΔΔCt^ method [38].

### 2.7. Mitochondrial DNA Content

The mitochondrial DNA copy number was determined through the ratio of mitochondrial DNA to nuclear DNA [39]. Sample DNA was diluted by using nuclease-free water to a final concentration of 5 ng/µL. To evaluate the relative copy number of mitochondrial DNA to nuclear DNA, the following genes were selected: *MT-ND1, MT-ND2,* and *MT-ND6* for mitochondrial genome; and *RPS11* and *RPP25* for nuclear encoded genes (Appendix A). SYBR Green quantitative PCR was prepared with 10 µL Precision Plus qPCR Master Mix (Primerdesign Ltd., Eastleigh, UK), 1 µL primer pairs (0.5 µM), 5 µL DNA (25 ng), and 4 µL nuclease-free water. Reactions were carried out in duplicate, using a StepOne Plus platform (Applied Biosystems, Waltham, MA, USA). Conditions of the qPCR: (preamplification step) 95 °C for 2 min, (amplification step) 40 cycles of 95 °C for 10 s, and 60 °C for 60 s. A melting curve analysis revealed a single amplicon in all cases. Analysis of mitochondrial DNA/nuclear DNA ratio was calculated by following the 2^−ΔΔCt^ method used for qRT-PCR analysis.

### 2.8. Plasma Analyte Determination

A Cobas Mira automated analyzer (Roche Diagnostics Ltd., Burgess Hill, UK) was used to measure levels of fasting glucose and plasma free fatty acids (FFAs), using commercially available assay kits (Alpha Laboratories Ltd., Eastleigh, UK) as per manufacturer’s instruction. Assay intra and inter-assay CV’s were <4% and <5%, respectively.

### 2.9. Hepatic Lipids Determination

A Triglyceride Determination Kit (TR0100, Sigma-Aldrich, Merck, Feltham, UK) was used to determine hepatic triglyceride content. Hepatic tissue was cut on dry ice, weighed, and homogenized in PBS. Samples were centrifuged for 30 s at 16,000× *g* (at room temperature), the lipid phase was transferred to a new tube, and all samples were assayed in duplicate, following the manufacturer’s instructions.

Hepatic total and free cholesterol were measured by using Cholesterol Quantification Assay kits (ab65359, Abcam, Cambridge, UK). Liver tissue was cut on dry ice and weighed. Lipids extracts were prepared by homogenizing samples in 200 μL of chloroform:isopropanol:NP-40 (7:11:0.1) preparation. Samples were centrifuged at room temperature for 10 min at 15,000× *g*; the liquid phase was transferred to a new tube, air dried at 50 °C to remove chloroform, and then placed under vacuum to remove trace organic solvent. Samples were dissolved by vortexing in 240 μL of Assay Buffer and assayed in duplicate, following the manufacturer’s instructions.

### 2.10. Hepatic Glycogen Determination

Hepatic glycogen content was measured by using Glycogen Assay Kit II (ab ab169558, Abcam, Cambridge, UK). Briefly, liver tissue was cut on dry ice, weighed, and homogenized with 200 μL distilled H_2_O on ice. Next, samples were boiled for 10 min and then centrifuged at room temperature for 10 min at 16,000× *g*. Insoluble material was removed, and all samples were assayed in duplicate, following the manufacturer’s instructions.

### 2.11. Hepatic ATP Determination

Hepatic ATP content was measured by using ATP Assay Kit (ab83355, Abcam, Cambridge, UK). Liver tissue was cut on dry ice, weighed, and homogenized with 100 μL ATP Assay Buffer on ice. Next, samples were centrifuged at 4 °C, at 13,000× *g*; then the insoluble material was removed, and all samples were assayed in duplicate, following manufacturer’s instructions.

### 2.12. Hepatic Reactive Oxygen Species (ROS) Determination

Hepatic ROS level was measured by using OxiSelect Assay Kit (STA-347, Cell Biolabs, Inc., San Diego, CA, USA). Liver tissue was cut on dry ice, weighed, resuspended in PBS at 50 mg/mL, and homogenized. Samples were centrifuged at room temperature for 5 min at 10,000× *g*, supernatants transferred to a new tube, and all samples were assayed in duplicate, following manufacturer’s instructions.

### 2.13. Hepatic DNA Damage Determination

Hepatic DNA damage was assessed by using DNA damage-AP sites-Assay Kit (ab211154, Abcam, Cambridge, UK). DNA was diluted to 50 μg/mL in TE buffer (10 mM Tris pH 7.5, 1 mM EDTA), and 5 μL of DNA was mixed with ARP solution and incubated at 37 °C for 1 h. Next, 90 μL of TE buffer, 1 μL of Glycogen Solution, and 10 μL of Sodium Acetate Solution were added to each sample and mixed. Then, 300 μL of absolute ethanol was added to each tube, and samples were vortexed briefly. Samples were incubated at −20 °C for 30 min and centrifuged at 4 °C for 20 min at 14,000× *g*. Supernatant was discarded, and DNA pellets were washed 3 times in 70% ethanol and air dried for 10 min. DNA pellets were dissolved in 10 μL of TE buffer, and DNA concentration was determined by using a NanoDrop 1000 spectrophotometer (Thermo Fisher Scientific, Wilmington, DE, USA). ARP-derived DNA samples were diluted to 1 μg/mL in TE buffer and assayed in duplicate, following the manufacturer’s instructions.

### 2.14. Hepatic Collagen Determination

Serial sections that were 5 μm thick were cut from the Bouin’s fixed paraffin blocks, using a Lecia RM2125RT microtome (Leica Microsystems Ltd., Milton Keynes, UK), then floated onto positively charged glass slides, two sections per slide. Slides were dried overnight at 42 °C. Prior to usage in downstream staining applications, slides were bathed in a series of graded concentrations of ethanol to deparaffinize and hydrate as follows: xylene twice, for five minutes each; 100% alcohol twice, for five minutes each; 90% ethanol once, for two minutes; and 70% ethanol once, for two minutes, before being washed in deionized water for five minutes to ensure excess ethanol was removed and tissues were fully rehydrated. To assess total collagen staining, nuclei was stained by submersion in Weigert’s haematoxylin for 8 min, before the slides were washed in a running tap water bath for 10 min. The slides were then stained for 1 h in a solution of Picro Sirius Red (0.5g Direct Red 80 (Sigma-Aldrich, Merck, Feltham, UK) in 500 mL of Picric acid solution (Sigma-Aldrich, Merck, Feltham, UK)). Slides were then washed twice in fresh acidified water (5 mL Glacial acetic acid (Sigma-Aldrich, Merck, Feltham, UK) in 1 liter of tap water). After shaking to remove water from the slides, slides were dehydrated by using 3 changes of 100% ethanol before a clearing in xylene and then mounted with coverslips, using Pertex (Vector Laboratories, Peterborough, UK).

To quantify Sirius Red staining of the liver tissues, sections were photographed at 20× magnification with an Olympus BH-2 light microscope (Olympus, Southend-on-Sea, UK) using CellSens Dimension software and examined blindly. Photos were uploaded to ImageJ (https://imagej.nih.gov/ij/download.html, accessed on 26 May 2022), and the instructions to quantify sections were followed. Briefly, the scale was changed to micrometers from the default pixel density before images were converted to grayscale. Red-stained collagen areas were segmented (isolated) by using thresholding. The thresholded area was measured. This returned a percentage area of staining. This process was executed in triplicate per section, creating a percentage area of staining for each. These were then averaged for each subject. Full instructions can be found here: https://imagej.nih.gov/ij/docs/examples/stained-sections/index.html (accessed on 26 May 2022).

### 2.15. Statistical Analysis

For single gene-expression analyses, the normality of all data sets was assessed with a Shapiro–Wilk test prior to further analysis, and if necessary, data were logarithmically transformed. For comparing means of two treatment groups with equal variances, unpaired, two-tailed Student’s *t*-test was performed, with *p* < 0.05 accepted as significant. RNAseq and proteomic outputs were analyzed by pairwise comparisons, with Benjamini and Hochberg false discovery rate control [37]. Correlation was calculated by using Pearson product-moment co-efficient. GraphPad Prism 8.0 software (GraphPad Prism Software, San Diego, CA, USA) was used to perform all statistical analysis, except RNAseq and proteomic analyses, where statistical analysis software package R (version 3.4.0) was utilized. Asterisks indicate level of significance based on the following criteria: * *p* < 0.05, ** *p* < 0.01, *** *p* < 0.001, and **** *p* < 0.0001.

## 3. Results

### 3.1. Altered Mitochondrial Function in Adolescent PA Males

The bioinformatic analysis predicted that adolescent PA males had mitochondrial dysfunction with decreased tricarboxylic acid cycle (TCA) potential, decreased oxidative phosphorylation (OXPHOS), and altered assembly of respiratory chain complex (Table 1).

Given the pivotal role of mitochondria in glucose and lipid metabolism, and oxidative stress response [32,40], we further examined oxidative phosphorylation in PA males, using ATP generation as an initial functional readout. ATP generation in the liver of PA males was significantly depressed (Figure 1A; *p* < 0.05). To determine if this was consequential of a decreased mitochondrial number, we examined mitochondrial DNA content in control and PA male offspring, determining the ratio of mitochondrial DNA to nuclear DNA [39]. There was no difference in the relative ratio of mitochondrial DNA copy number of *MT-ND1*, *MT-ND2*, and *MT-ND6* between adolescent control and PA males (Figure 1B), indicative of there being no significant alteration of mitochondrial numbers. We therefore interrogated RNAseq and proteomic data sets from the same animals to determine the mechanistic underpinnings of this decreased ATP generation.

### 3.2. Decreased Oxidative Phosphorylation in Adolescent PA Males

**TCA cycle:** Adolescent PA males had transcript and protein profiles indicative of reduced NADH and FADH2 production capacity, further suggesting impacts upon oxidative phosphorylation. (Figure 2A; Appendix A).

**OXPHOS:** (Figure 2B; Appendix A). Adolescent PA males had 13 significantly decreased genes in complex I, with 11 further genes demonstrating a pattern of shift towards decreased expression. In complex III there were 3 significantly decreased genes and 4 genes with a trend towards decreased expression in adolescent PA males. In complex IV there were 4 genes whose expression was decreased as compared to control male samples and 6 genes with a trend towards decreased expression. Finally, in complex V adolescent PA males had 5 significantly decreased genes and a trend for decreased expression of another 10 genes.

Collectively, these data point strongly toward decreased oxidative phosphorylation activity, functionally culminating in decreased ATP generation in the liver of PA males. In further support of this explanation, we also observed positive correlations between expression OXPHOS components and hepatic ATP levels in control and PA animals (Appendix A). To comprehend the aetiology of these observations, we examined selected hepatic OXPHOS gene expression in fetal life. Prenatal androgenization in the male fetus did not influence the transcript abundance of *NDUFA7, NDUFA12,* and *NDUFB4* (Complex I); *CYCS, UQCRB,* and *UQCRC1* (Complex III); *COX5A, COX6B1,* and *COX7B* (Complex IV); and *ATP5IF1, ATP5MC3,* and *ATP5PO* (Complex V) (Figure 3). This suggested that reprogramming of OXPHOS evident in fetal life as a direct consequence of prenatal androgen excess was not permanent. Thus, we next examined substrate availability in postnatal life.

### 3.3. Increased FA Uptake into Mitochondria

Although there were increased circulating FFAs (Figure 4A) and trends toward increased expression of hepatic fatty acid transport protein SLC27A5 (q = 0.12; *p* < 0.01) and fatty acid binding protein *FABP5* (q = 0.19; *p* < 0.05) in PA males, we observed no strong evidence to indicate increased FA uptake potential into the liver (Appendix A), and there was no increase in hepatic triglycerides (Figure 4B). However, PA males expressed increased hepatic mRNA encoding genes facilitating the entry of fatty acids into mitochondria, *CPT1A* and *CPT1B* (q < 0.05), and tended toward increased *CPT2* (q = 0.09; *p* < 0.01) and *CRAT* (q = 0.09; *p* < 0.01). Concurrent with increased mitochondrial uptake of fatty acids potential, there was evidence of decreased fatty acid synthesis in PA males, with decreased expression of *ACLY*, *ACACA*, and *FASN* (q < 0.05) and a trend toward increased *MLYCD* (q = 0.06; *p* < 0.01) (Appendix A). In further support of these observations, PA males had a decreased expression of *ELOVL6* (q < 0.05), which is involved in fatty acid elongation, and decreased expression of fatty acid desaturase *SCD* (q < 0.05) (Appendix A).

### 3.4. Reduced Mitochondrial FA Utilization

Whilst there was an increased capacity for mitochondrial FA uptake, there was downstream evidence for reduced utilization of FAs by mitochondria (Appendix A). PA males had a decreased expression of fatty acyl-CoA synthetases facilitating activation of fatty acids prior to oxidation, i.e., ACSS2 and ACSL3 (q < 0.05); and a trend toward decreased expression of ACSL5 (q = 0.11; *p* < 0.05). Furthermore, PA males had decreased expression of short/branched-chain acyl-CoA dehydrogenase ACADSB (q < 0.05) and likely decreased expression of ECHS1 (q = 0.09; *p* < 0.01) and DECR1 (q = 0.06; *p* < 0.01), enzymes of mitochondrial beta oxidation pathway, preferentially oxidizing short-, medium-, and long-chain fatty acids. This was not the case in peroxisomes, as key peroxisomal pathways were increased (Appendix A). PA males had a trend for increased protein concentrations of ACOX2 (q = 0.08; *p* < 0.001), EHHADH (q = 0.1; *p* < 0.01), and ACAA1 (q = 0.07; *p* < 0.001), enzymes of peroxisomal beta oxidation. Collectively, these data suggest increased mitochondrial FA uptake without concomitantly increased mitochondrial beta oxidation in adolescent PA males.

### 3.5. Reduced Mitochondrial Glucose Utilization

Whilst here was no difference in plasma glucose levels (Figure 4C) and no difference in glycolysis-related gene expression between PA males and control animals (Appendix A), we did observe an increased expression of *PC* (q < 0.05) and trend toward increased expression of *FBP2* (q = 0.09; *p* < 0.05), genes encoding enzymes involved in gluconeogenesis. Adolescent PA males had increased hepatic glycogen content (Figure 4D; *p* < 0.05) likely underpinned by observed increased expression of *GYS2* (q = 0.11; *p* < 0.05), a rate limiting step in glycogen synthesis.

The gene and protein expressions of components of the pyruvate dehydrogenase complex were reduced, e.g., *DLAT* (q < 0.05), and there were trends toward decreased protein concentrations of PDHA1 (q = 0.08; *p* < 0.001). PA males had also increased gene expression of *PDK4* kinase (q < 0.05), which phosphorylates and inhibits activity of pyruvate dehydrogenase complex, further underscoring the likelihood of decreased pyruvate dehydrogenation (Appendix A). Collectively, these observations are indictive of decreased hepatic glucose utilization and increased hepatic glucose storage in PA males.

### 3.6. Hepatic free Cholesterol Correlates with Mitochondrial Dysfunction

The lack of utilization of substrate that is abundantly available suggested an integral mitochondrial problem. Increased cholesterol can impact mitochondrial membrane fluidity, membrane proteins, and transporters [41]. PA males had increased an expression of hepatic LDL receptor–related proteins *LRP1* and *LRP5*, mediating uptake of chylomicron remnants [23], and a strong trend toward the decreased expression of *NPC1* (q = 0.06; *p* < 0.01), which is involved in removal and trafficking of cholesterol from late endosomes to other organelles. Additionally, we noted an increased expression of *SOAT2* (q < 0.05), which is involved in cholesterol ester synthesis; an increased mRNA expression encoding lipid droplet protein *PLIN5* (q < 0.05); and a trend toward an increased expression of *PLIN2* (q = 0.07; *p* < 0.01) (Appendix A). As a functional readout of this altered gene expression profile, PA males had increased hepatic concentrations of total cholesterol (Figure 4E; *p* < 0.05) and hepatic free cholesterol (Figure 4F; *p* < 0.05). Hepatic free cholesterol correlated with altered expression of OXPHOS genes (Appendix A). Considering increased hepatic cholesterol accumulation in adolescent PA males, we investigated mRNA levels of mitochondrial transporters, translocases, and channels. Adolescent PA males had an altered expression of several genes from mitochondrial solute carrier family, inner and outer mitochondrial translocases, and ionic channels (Appendix A).

### 3.7. Reduced Capacity for Mitochondrial Rejuvenation

In terms of putative mitochondrial dysfunction in the livers of PA males, we next examined mitochondrial dynamics/turnover (fission and fusion) and “quality control” pathways. Gene expression associated with mitochondrial fusion was unaffected by prenatal androgen excess (Appendix A); however, PA males had a decreased expression of *DNM1L* (q < 0.05), the key regulator of mitochondrial fragmentation (fission). Furthermore, PA males had a reduced expression of *MTFP1* (q < 0.05) and trend toward decreased expression of *MTFR1* (q = 0.07; *p* < 0.01), genes involved in mitochondrial fission. In addition, PA males had an altered capacity for mitophagy. There was an increased expression of *USP30* (q < 0.05); a trend toward increased *PARK2* (q = 0.07; *p* < 0.01) and *ZNF746* (q = 0.08; *p* < 0.01); a decreased expression of *CALCOCO2*, *FUNDC*, *GABARAPL2*, *PHB1*, and *PARK7* (q < 0.05); and a trend toward decreased expression of *OPTN* (q = 0.09; *p* < 0.01), *PINK1* (q = 0.19; *p* < 0.05), and *PGAM5* (q = 0.07; *p* < 0.01). This then provided further evidence of mitochondrial dysfunction at a level of reduced capacity to cope with mitochondrial damage and dysfunction in PA males as compared to contemporary control animals in adolescence.

### 3.8. Increased Hepatic ROS Content in Adolescent PA Males

Dysfunctional mitochondria are an important source of reactive oxygen species (ROS) that can promote the progression of NAFLD and activation of profibrogenic mechanisms. In line with previous indications of decreased antioxidant capacity [23], PA males had an increased hepatic ROS content (Figure 5A; *p* < 0.05), which negatively correlated with antioxidant potential (Appendix A). Importantly, this was accompanied by a compelling trend toward increased hepatic DNA damage in adolescent PA males as compared with controls (Figure 5B; *p* = 0.059). There was a positive correlation of ROS with the regulation of lipid metabolism and mitochondrial fatty acid uptake (Appendix A).

### 3.9. Increased Hepatic Collagen Deposition in PA Males

As reported previously, PA males had an increased expression of hepatic genes and proteins relevant to liver fibrosis [23]. These are positively correlated with ROS content (Appendix A). To further evaluate hepatic health in PA males, we assessed collagen deposition by using Sirius Red staining in liver sections from control and PA males. Our assessment of liver biopsies indicated increased liver fibrosis in PA males, with significantly increased hepatic collagen accumulation (2.9 ± 1.2%), as compared to the control samples (1.1 ± 0.38%) (Figure 5D; *p* < 0.0001), and this is indicative of PA animals already displaying the early stages of liver damage.

## 4. Discussion

Males have a higher prevalence and severity of NAFLD as compared to females [42], and in females, increased androgen levels are associated with an elevated risk of NAFLD development and progression [30,43]. In animal models, females exposed to increased levels of androgens in utero develop NAFLD in adult life, indicating that androgens might be potential drivers of NAFLD development [44,45]. However, the effect of prenatal androgen overexposure on males has not been studied. Sons of hyperandrogeanemic PCOS women have elevated circulating cholesterol and triglycerides [16,17]. Similarly, male sheep overexposed to androgens in fetal life develop dyslipidaemia in adolescence [23]. Dyslipidaemia and altered lipid metabolism are often associated with NAFLD development [28,29]. Here, we investigated whether the excess developmental androgen exposure in males affects hepatic lipid metabolism and mitochondrial function, using our outbred clinically realistic large-animal model, previously shown to recreate the hormonal profile noted in first degree male relatives of PCOS sufferers. Adolescent PA males had increased circulating FA and hepatic cholesterol accumulation; altered glucose and FA metabolism; hepatic mitochondrial dysfunction, with altered mitochondrial transport; decreased TCA, OXPHOS, and ATP synthesis; and impaired mitophagy. Mitochondrial dysfunction in PA males was further associated with increased hepatic ROS level and signs of early liver fibrosis.

Elevated circulating FAs are often associated with metabolic syndrome [46] and NAFLD development [47], likely due to increased hepatic uptake of FA [48], which is proportional to FA blood concentration [49]. PA males had increased circulating FA. There was no difference in the expression of FA transporters in PA males as compared with controls; however, there was significantly increased availability of FA for transport from circulation into the liver. In the liver, FAs undergo esterification into triglycerides or are oxidized. There was no evidence of increased FA esterification in PA males, as confirmed by unaltered TG concentrations. Interestingly, FA incorporation into TG can be a protective mechanism, as the accumulation of intracellular FA or lipid metabolites in the liver can cause insulin resistance, liver injury, and dysfunction [50,51]. Therefore, in view of the likely increased circulating FA and, thus, hepatic FA delivery in PA males, a lack of attendant increase in TG synthesis may have negative health consequences.

PA males had an increased expression of *PPARA* [23], an upstream regulator of FA oxidation [52], and an increased expression of genes and proteins facilitating mitochondrial FA uptake, a rate-limiting step in beta-oxidation [53], a major route of hepatic FA disposal [54], potentially a mechanistic attempt to restore hepatic lipid homeostasis. However, increased mitochondrial FA uptake in PA males was not accompanied by the upregulation of mitochondrial beta-oxidation pathway that we suggest may lead to mitochondrial FA overload and subsequent mitochondrial dysfunction [55]. PA males had, however, a trend for increased protein level of enzymes of peroxisomal oxidation and a trend for increased expression of *CROT*, transporting shortened FA out of the peroxisomes for complete oxidation in mitochondria [56]. Increased peroxisomal FA oxidation in PA males might be a compensatory response that aims to prevent hepatic lipid accumulation; however, peroxisomal oxidation generates high levels of ROS [57], and this is in agreement with elevated ROS levels in PA males.

PA males had increased PDH kinase 4 (*PDK4*) and decreased the expression of components of PDH complex, thus indicating decreased glucose utilization [58]. Increased hepatic glucose concentrations stimulate glycogen synthase (GYS2), a key enzyme in glycogen synthesis [59]. In keeping with this, PA males displayed a trend toward elevated *GYS2* expression and, functionally, displayed increased accumulation of hepatic glycogen; this is in line with reports of hepatic carbohydrate metabolism dysregulation in NAFLD resulting in hepatic glycogen accumulation [60]. Glycogenosis, excess hepatocyte glycogen, is common in adult and paediatric NAFLD and is associated with liver cell injury, megamitochondria, and NASH, but reduced steatosis grades [61].

Hepatic steatosis is characterized by excessive accumulation of triglycerides and cholesterol esters, stored in lipid droplets [62]. The accumulation of lipids in the liver may lead to lipotoxicity, oxidative stress, lipid peroxidation, inflammation, and mitochondrial dysfunction, priming the liver for what is often irreversible injury [63]. Adolescent PA males had normal hepatic triglyceride concentrations but increased hepatic concentrations of total and free cholesterol, as compared with controls, and increased expression of lipid droplet proteins *PLIN2* and *PLIN5*, indicating the potential for increased lipid droplet accumulation [64,65]. PLIN2 is increased in human NAFLD and correlates with inflammation, fibrosis, and oxidative damage reflected in ballooned hepatocytes [66]—importantly, PLIN2 also inhibits FA oxidation [67]. In a rodent model of NASH, hepatic PLIN2 ablation protected against hepatic steatosis and inflammation [68]. PLIN5 promotes lipid storage [69], and it is also increased in human steatosis [70]. Triglycerides mainly represent a “safe” storage for FA [50], whereas the accumulation of other lipotoxic lipids, such as cholesterol, FA, diacylglycerol, and ceramides, leads to cellular dysfunction [71,72]. Cholesterol accumulates in the liver when the total sum of the pathways involved in cholesterol synthesis and uptake exceeds the pathways involved in cholesterol removal [71]. PA males had elevated plasma cholesterol and increased expression of hepatic LDL receptor–related proteins *LRP1* and *LRP5*, mediating the uptake of chylomicron remnants [23]. In addition, there was a decreased expression of hepatic *CYP7A1* in PA males [23], encoding the rate-limiting enzyme in the classical pathway converting cholesterol to bile acids, also likely contributing to increased hepatic cholesterol accumulation in those males.

Hepatic cholesterol accumulation is an important aetiological factor underlying progression from NAFLD to fibrosing NASH, both in animal models and humans [71,73]. Increased hepatic cholesterol leads to dysfunction of many hepatocyte organelles and activation of other liver cells central to fibrosis in NASH [74]. The accumulation of free cholesterol affects the fluidity of cell membranes and function of transmembrane proteins [75]. Cholesterol may be oxidized to oxysterols that are involved in NAFLD-associated liver injury [76]. Mitochondria are sensitive to such changes in cholesterol concentrations, due to low basal membrane cholesterol content as compared to other cellular membranes [77]. Reduced mitochondrial membrane fluidity, due to increased cholesterol content, impacts the function of membrane proteins and transporters, resulting in the depletion of the mitochondrial glutathione pool [78]. Increased cholesterol also sensitizes mitochondria to oxidative stress, leading to mitochondrial insufficiency and metabolic abnormality [77,79]. Murine studies demonstrated that increased hepatic cholesterol results in elevated cholesterol in mitochondrial membranes, disrupted assembly of mitochondrial respiratory complexes, decreased OXPHOS, reduced ATP, altered mitochondrial dynamics and mitochondrial morphology, depleted cellular and mitochondrial antioxidant potential, and increased generation of ROS and signs of liver injury [80,81,82,83], reminiscent of the outcomes of prenatal androgen excess reported here. Similarly, mitochondrial dysfunction and decreased ATP synthesis have been reported in patients with NAFLD and NASH [84,85,86,87,88,89].

Adolescent PA males had an altered expression of genes and proteins involved in mitochondrial fission and mitophagy. Mitochondrial turnover is an integral aspect of quality control in which dysfunctional mitochondria are selectively eliminated through mitophagy and replaced through expansion of pre-existing mitochondria [63]. Impaired mitochondrial quality control results in the accumulation of damaged mitochondria that affect its function. Adolescent PA males had decreased the expression of genes involved in mitochondrial fission, i.e., *DNM1L* and *MTFP1*, and trend toward a decreased expression of *MTFR1*. DNM1L is a key regulator of mitochondrial fission, and depletion of DNM1L, MTFP1, and MTFR1 is associated with the formation of megamitochondria, decreased mitochondrial fragmentation, reduced mitochondrial respiration coupled with diminished cellular ATP, and increased ROS level [90,91,92,93]. Mitophagy is regulated through receptors on mitochondrial outer membrane or ubiquitin molecules linked to mitochondrial surface proteins, resulting in the formation of autophagosomes surrounding mitochondria. PA males had a trend toward increased expression of *PARK2*, encoding PARKIN, an E3 ligase recruited to damaged mitochondria to initiate ubiquitination of mitochondrial outer membrane proteins, and subsequent mitochondrial degradation by mitophagy [94]. However, regardless of PARKIN expression, PA males also had a trend for decreased expression of *PINK1* and reduced expression level of *PARK7*, mitophagy inducers and regulators, promoting ubiquitylation and proteasomal degradation of unfolded or misfolded parkin substrates [95]. PINK1 is an upstream regulator critical for recruiting and activating PARKIN [96]. The loss of PINK1 results in mitochondrial dysfunction, decreased OXPHOS, and increased sensitivity to oxidative stress [97,98]. In addition, PA males had an increased expression of *USP30*, which removes ubiquitin from damaged mitochondria and blocks parkin-mediated mitophagy [99]. Coupled with reduced expression of *CALCOCO2*, *OPTN*, and *FUNDC1*, a picture of disrupted mitochondrial dynamics and opportunity for increased presence of damaged mitochondria in PA males emerges [100,101]. We also note a trend toward an increased expression of PARIS, encoded by *ZNF746*, a repressor of PGC-1α [102], the master regulator of mitochondrial biogenesis [103]. Thus, we propose that dysfunctional mitophagy in PA males is a consequence of prenatal androgen excess, and, in this regard, it is pertinent to note that dysfunctional mitophagy is associated with NAFLD progression, whilst increased mitochondrial autophagy can attenuate fatty liver disease [86,104,105,106,107,108].

Early stages of NAFLD are characterized by mitochondrial flexibility, where mitochondria increase their activity and number to alleviate hepatic lipid overload [63,81]. This hepatic mitochondrial plasticity is lost during the transition from NAFLD to NASH in both animal models and humans, with diminished OXPHOS, decreased ATP, mitochondrial structural defects, increased ROS synthesis, reduced antioxidant potential, and oxidative DNA damage [81,86,89,109]. The significance of prenatal androgen excess to drive a hepatic dysfunctional mitochondria state is wide reaching. The liver requires energy in the form of ATP, mainly originating from OXPHOS, and is, hence, a mitochondria-rich organ [110], as such mitochondria constitute ~18% of the entire hepatocyte volume [111]. Mitochondrial dysfunction affects not only energy production but also disturbs many vital cellular functions and causes hepatocyte damage, ultimately initiating or exacerbating many disorders [110]. Decreased ATP affects the mitochondrial ability to maintain structure and function. Oxidative stress associated with mitochondria dysfunction, resulting from cholesterol or FA overload, significantly contributes to hepatocyte damage, inflammation, and progression of NAFLD to NASH [110]. Mitochondria, together with peroxisomes and endoplasmic reticulum, are significant contributors to the cellular ROS pool [112]. Mitochondrial superoxide dismutase (SOD), together with glutathione, glutaredoxin, and thioredoxin systems, is responsible for limiting ROS accumulation [110]. We have previously reported that adolescent PA males had decreased expression of multiple components of antioxidant system [23]. Combined with the decreased expression of *SLC25A11*, a mitochondrial glutathione carrier [113], this discrepancy between increased ROS level, as measured here, and reduced antioxidant defenses in PA males will promote oxidative stress. In combination with hepatic cholesterol accumulation, mitochondrial dysfunction, and reduced ATP, increased oxidative stress could, in turn, promote the development of NASH and fibrosis. In support of this suggestion, we noted herein evidence of elevated hepatic collagen deposition, together with previously reported increased expression of hepatic collagen genes and fibrosis drivers, and increased collagen plasma proteins in PA males [23]. NAFLD/NASH patients and animal models have impaired antioxidant potential and increased ROS generation, resulting in a pro-oxidant state [114,115] that is strikingly similar to our findings in adolescent PA males. We noted a trend for increased DNA damage in PA males. Oxidative DNA damage has been reported in NAFLD, with NASH patients having higher levels of oxidative DNA damage compared to other liver disorders [116,117]. In mitochondria, ROS react primarily with iron–sulfur cluster-containing proteins, including proteins in TCA cycle and OXPHOS [110,118]. In addition, the PDH complex is also susceptible to redox reactions [110]. This agrees with the observed decreased expression of genes and proteins involved in the TCA cycle, OXPHOS complexes, and PDH complex in PA males. Interference with these proteins establishes a self-perpetuating vicious cycle of sustained ROS release, therefore further harming liver tissue, impairing mitochondrial activity, and, thus, exacerbating disease progression.

NAFLD is a risk factor for cardiovascular disease (CVD), especially coronary heart disease, but also arrhythmia, stroke, and heart failure, independent of traditional risk factors, such as obesity, diabetes, hypertension, and dyslipidaemia [119,120]. Although all stages of NAFLD are linked with an elevated risk of CVD, those with NASH and fibrosis are especially at risk [119,120]. Microvascular damage and endothelial dysfunction, as a result of increased oxidative stress and persistent inflammation, altered lipid metabolism, dyslipidaemia, insulin resistance, synthesis of procoagulant factors, and increased vascular remodeling, have been highlighted as potential underlying mechanisms of increased CVD risk in NAFLD [119,120]. Given the combination of factors reported here and in our previous study [23], it can be speculated that these ovine PA males are at increased risk of CVD; however, whether or not similar risk alteration occurs in human males from PCOS pregnancies remains currently unknown.

The main limitation of our study is that only a single postnatal time point was studied. Nonetheless, we have no reason to consider that the impairments noted here would be likely to spontaneously resolve themselves, and, in this regard, it is of concern that such dysfunction is already evident as early as adolescence. We also acknowledge species differences between humans and sheep; however, our data closely parallel human clinical outcomes of hyperandrogenaemic pregnancies such as those in PCOS. As there is no effective treatment for NASH with advanced fibrosis, this preliminary animal modeling provides the first indication that males from hyperandrogenaemic pregnancies should be considered to be at an increased risk of serious hepatic disease; thus they would benefit from consideration in terms of investigation and intervention to mitigate against disease progression.

## 5. Conclusions

In conclusion, prenatal androgen overexposure in males drives the increased accumulation of hepatic cholesterol and glycogen, together with perturbed glucose and FA metabolism, mitochondrial dysfunction, increased hepatic ROS accumulation, and early life liver fibrotic deposition. Taken together these findings are directly clinically relevant to our mechanistic understanding of NASH development, independent of diet or obesity.

## Figures and Tables

**Figure 1 biomedicines-10-01291-f001:**
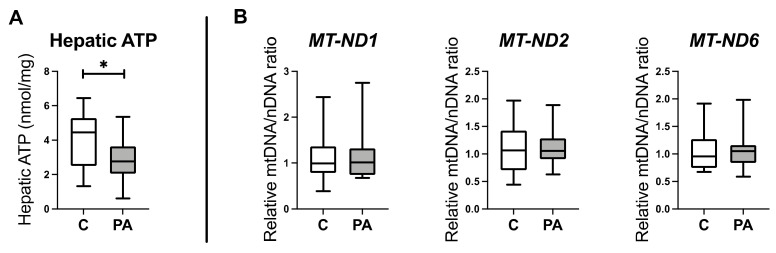
Prenatally androgen-exposed males have a decreased hepatic ATP level that is independent of mitochondria number. (**A**) Hepatic ATP level in adolescent prenatally androgen-exposed males (PA; *n* = 14) and control males (C; *n* = 14). (**B**) Relative copy number of mitochondrial DNA to nuclear DNA in adolescent prenatally androgen-exposed males (PA; *n* = 14) and control males (C; *n* = 14). Differences were analyzed by unpaired, two-tailed *t*-test. (* *p* < 0.05). Box-plot whiskers are lowest and highest observed values, and box is the upper and lower quartile, with median represented by line in box.

**Figure 2 biomedicines-10-01291-f002:**
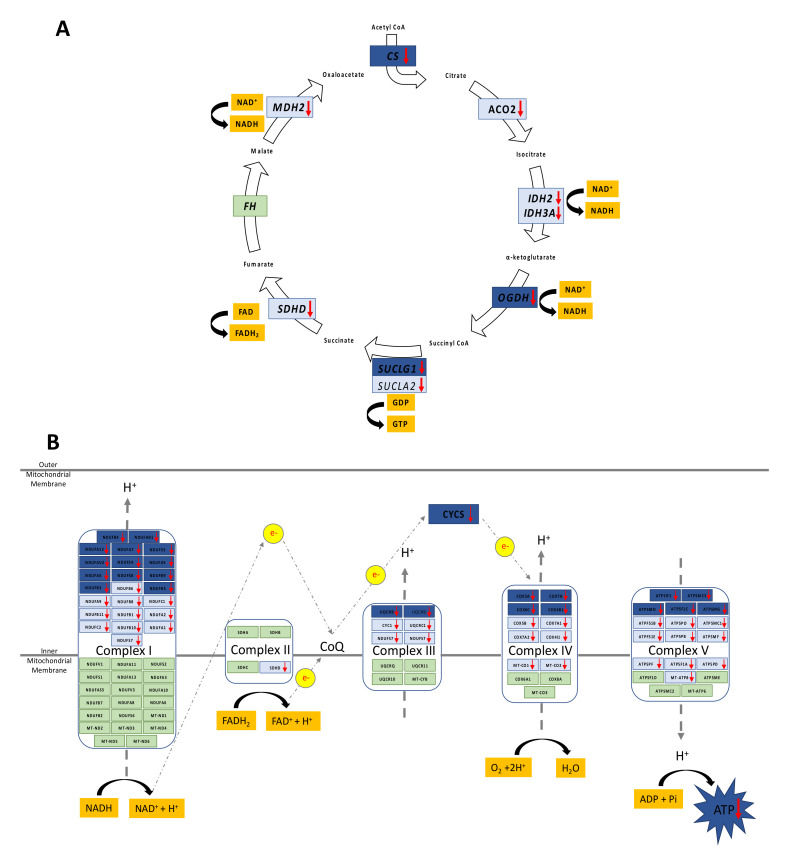
Prenatally androgen-exposed males have decraesed TCA and OXPHOS cycles. Schematic representation of (**A**) TCA and (**B**) OXPHOS cycles in adolescent prenatally androgen-exposed males. In dark blue boxes, significantly decreased genes as compared with control males with q < 0.05. In light blue boxes, trend for decreased gene expression with q < 0.2. In green boxes, genes with no difference in expression between control and PA males. RNAseq data were examined by pairwise comparisons with false discovery rate (q-value) determined by Benjamini–Hochberg method.

**Figure 3 biomedicines-10-01291-f003:**
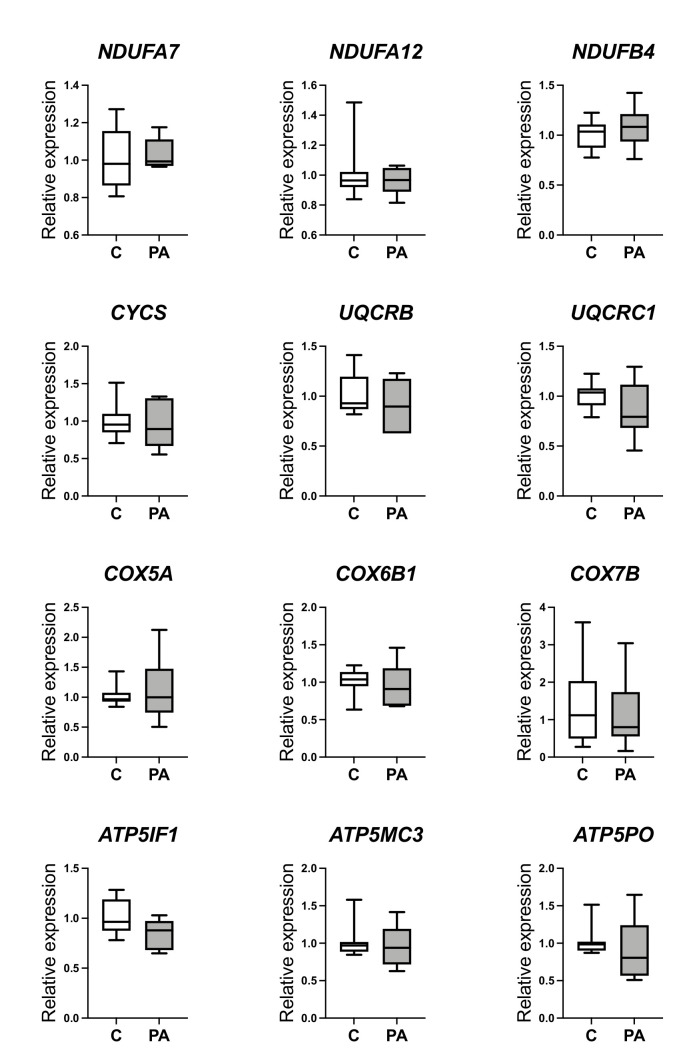
Decreased expression of OXPHOS genes is not evident in fetal life. Expression of selected hepatic OXPHOS genes, identified as differentially expressed during adolescence, was examined by using qRT-PCR in control (C; *n* = 10) and prenatal androgen excess (PA; *n* = 6) male fetal livers (day 90 of gestation). Differences were analyzed by unpaired, two-tailed *t*-test. Box-plot whiskers are lowest and highest observed.

**Figure 4 biomedicines-10-01291-f004:**
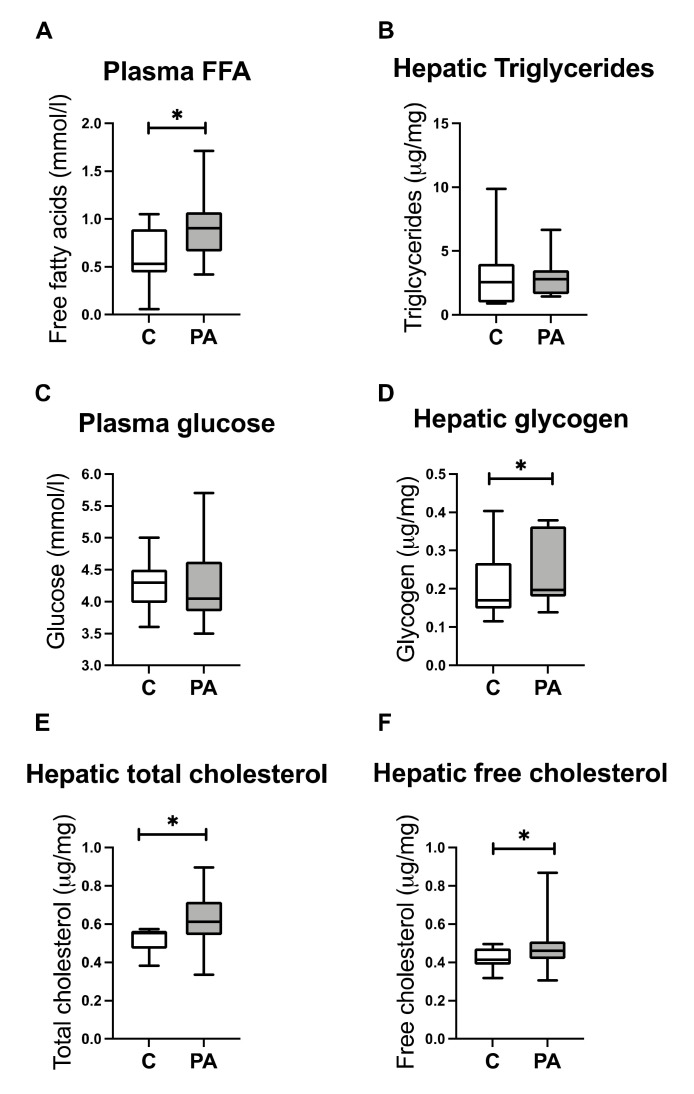
Prenatally androgen-exposed male offspring have increased plasma free fatty acids and develop increased hepatic glycogen and total and free cholesterol accumulation. Plasma and liver samples were collected at 6 months postnatal age from control (C; *n* = 14) and prenatal androgen excess (PA; *n* = 14) male offspring. Plasma free fatty acids and glucose level in adolescent control and PA males (**A**,**C**). Level of hepatic triglycerides, glycogen, total and free cholesterol in adolescent control and PA males (**B**,**D**–**F**). Statistical testing by unpaired, two-tailed Student’s *t*-test (* *p* < 0.05). Box-plot whiskers are lowest and highest observed values, and box is the upper and lower quartile, with median represented by line in box.

**Figure 5 biomedicines-10-01291-f005:**
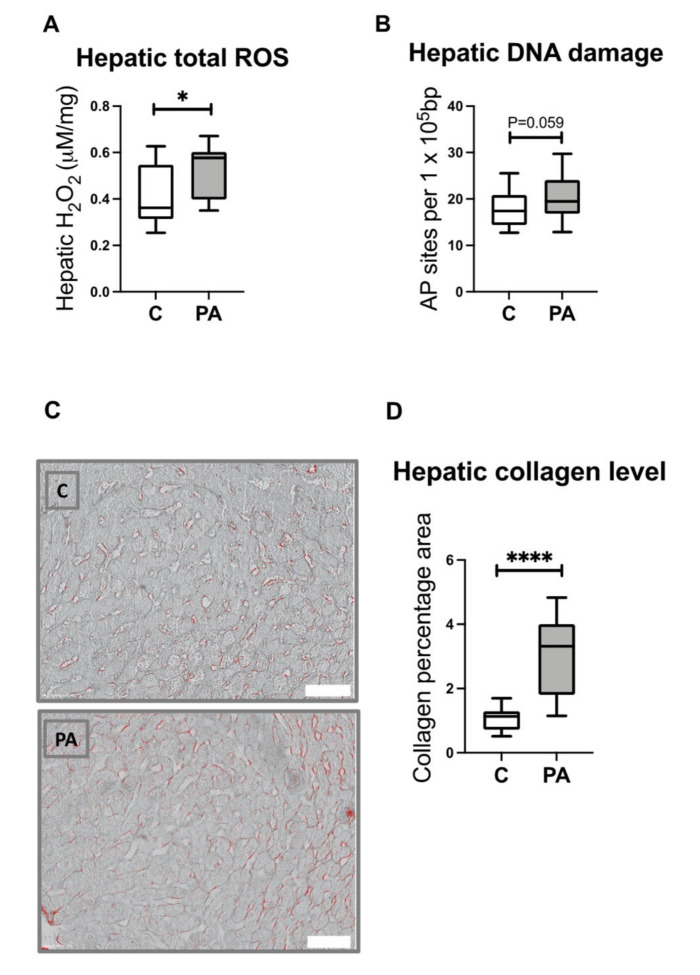
Prenatally androgen-exposed male offspring have increased hepatic ROS and collagen levels. (**A**,**B**) Liver samples from control (C; *n* = 14) and prenatal androgen excess (PA; *n* = 14) were collected at 6 months postnatal age from male offspring. ROS and DNA damage were assessed by using commercially available assays. (**C**) Representative liver sections from control and PA males stained with Sirius Red. Scale bars (white rectangles) represent 200 μm. (**D**) Sirius Red staining was quantified by using digitized ImageJ analysis. Differences were analyzed by unpaired, two-tailed *t*-test (* *p* < 0.05; **** *p* < 0.0001). Box plot whiskers are lowest and highest observed values; box is the upper and lower quartile, with median represented by line in box.

**Table 1 biomedicines-10-01291-t001:** Bioinformatic characterization of adolescent PA males.

Pathway	*p*-Value	Direction	Molecules
Mitochondrial Function			
Mitochondrial dysfunction	5.43 × 10^−11^	NP	31 genes
	2.85 × 10^−12^	NP	20 proteins
Oxidative phosphorylation	3.75 × 10^−9^	↓↓	22 genes
	1.02 × 10^−7^	↓↓	11 proteins
TCA cycle	1.13 × 10^−7^	↓	8 proteins
Mitochondrial respiratory chain deficiency	3.36 × 10^−7^	NP	10 proteins
Assembly of respiratory chain complex	3.37 × 10^−4^	NP	5 proteins

Dysregulated hepatic mitochondrial function associated with differentially expressed genes and proteins in adolescent PA males; ↓ = mild directionality prediction (Z score ≤ 2/−2); ↓↓ = strong directionality prediction (Z score > 2/−2); NP = no prediction made.

## Data Availability

Raw RNA sequencing data are available from the ArrayExpress database (http://www.ebi.ac.uk/arrayexpress; last accessed 26 May 2022) under accession number E-MTAB-8032. Mass spectrometry proteomics data were deposited to the ProteomeXchange Consortium via the PRIDE140 partner repository with the dataset identifier PXD014050.

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
