# Peer review of "Hepatic Mitochondrial Dysfunction and Risk of Liver Disease in an Ovine Model of “PCOS Males”"

_biomedicines, 2022, doi:10.3390/biomedicines10061291_

Round 1
Reviewer 1 Report
Dear authors,
with exhaustive analysis you determined in an animal model the hepatic mitochondrial dysfunction and risk of liver disease in so called ‘PCOS males’, male offsrping in sheep that have been treated by testosteron in utero to mimick PCOS driven effect during pregnancy. Liver RNA sequencing and proteome analysis guided your experiments and supportive analyses have been performed to sustain observed effects on mitochondrial dysfunction.
For the reviewer the work done is sufficiently analysed and "backed up" by additional experiments so that the results are reliable and accountable.
As usually proteome and transcriptome analysis gains a lot of data, but presentation and further analyses sustain basic finding in an elegant way.
Minor spelling errors have been deteted in lines 386 (double "had") and in line 467 where a dot is missing after sufferers.
Author Response
We would like to thank the reviewer for their kind words, and their expertise, time and effort in reviewing our manuscript – it is very much appreciated by us. We have corrected the typographical errors identified.
Reviewer 2 Report
The study is suited for the journal. The argument is interesting, written in a correct form and overall reading is feasible. The study investigates whether excess developmental androgen exposure in large-animal model affects hepatic lipid metabolism and mitochondrial function.
Before publication major focal points must be addressed.
- The title is misleading it should be something like: "Hepatic mitochondrial dysfunction and risk of liver disease in a ovine model of ‘PCOS males’ ".
- The arguments could be augmented with a view on cardiovascular disease as metabolic shift results in increased CV burden.
- The study would have greatly benefited from liver ultrasound assessment. This is essential for clinical translation.
- Oxidative stress should have been highlighted also in hepatic endothelial disfunction.
- In the conclusions, future prospectives, translational and treatment considerations are missing.
Author Response
The study is suited for the journal. The argument is interesting, written in a correct form and overall reading is feasible. The study investigates whether excess developmental androgen exposure in large-animal model affects hepatic lipid metabolism and mitochondrial function.
We would like to thank the reviewer for their kind words, and their expertise, time and effort in reviewing our manuscript – it is very much appreciated by us.
Before publication major focal points must be addressed.
The title is misleading it should be something like: "Hepatic mitochondrial dysfunction and risk of liver disease in a ovine model of ‘PCOS males’ ".
We have now changed the title as suggested by the reviewer.
The arguments could be augmented with a view on cardiovascular disease as metabolic shift results in increased CV burden. Oxidative stress should have been highlighted also in hepatic endothelial disfunction.
We have now added a section discussing potential cardiovascular consequences of NAFLD and oxidative stress in prenatally androgenized males.
The study would have greatly benefited from liver ultrasound assessment. This is essential for clinical translation.
This is a very good suggestion, unfortunately, this is not possible as the animals have been sacrificed already. We did not know, during study conduct, that this analysis would be warranted, but will factor into future studies.
In the conclusions, future prospectives, translational and treatment considerations are missing.
We believe that this study is only preliminary and therefore discussing future prospective and treatment considerations at this stage would be too far reaching. Whilst tempting to speculate, this would be pure speculation and would therefore not be suitable for the conclusions section. Hence we have not included this in our revised manuscript.
Round 2
Reviewer 2 Report
The authors have faced the highlighted issues according to suggestions. The article is now suitable for publication.